# Nurse home visiting to improve child and maternal outcomes: 5-year follow-up of an Australian randomised controlled trial

Sharon Goldfeld[1,2,3]*, Hannah Bryson[1,2], Fiona Mensah[2,3], Anna Price[1,2,3], Lisa Gold[4], Francesca Orsini[5,6], Bridget Kenny[1,2], Susan Perlen[1,2], Shalika Bohingamu Mudiyanselage[4], Penelope Dakin[7], Tracey Bruce[8], Diana Harris[7], Lynn Kemp[8]

**1** Centre for Community Child Health, The Royal Children's Hospital, Parkville, VIC, Australia, **2** Population Health, Murdoch Children's Research Institute, Parkville, VIC, Australia, **3** Department of Paediatrics, University of Melbourne, Parkville, VIC, Australia, **4** School of Health and Social Development, Deakin University, Geelong, VIC, Australia, **5** Clinical Epidemiology and Biostatistics Unit, Murdoch Children's Research Institute, The Royal Children's Hospital, Parkville, VIC, Australia, **6** Melbourne Children's Trials Centre, Murdoch Children's Research Institute, The Royal Children's Hospital, Parkville, VIC, Australia, **7** Australian Research Alliance for Children and Youth, Canberra City, ACT, Australia, **8** Ingham Institute, Western Sydney University, Penrith, NSW, Australia

* sharon.goldfeld@rch.org.au

**Data Availability Statement:** Data are fully available to researchers who request access to the data from the Melbourne Children's Campus LifeCourse institutional data access platform (https://lifecourse.melbournechildrens.com/data-

## Abstract

### Objectives

Nurse home visiting (NHV) is widely implemented to address inequities in child and maternal health. However, few studies have examined longer-term effectiveness or delivery within universal healthcare systems. We evaluated the benefits of an Australian NHV program ("right@home") in promoting children's language and learning, general and mental health, maternal mental health and wellbeing, parenting and family relationships, at child ages 4 and 5 years.

### Setting and participants

Randomised controlled trial of NHV delivered via universal, child and family health services (the comparator). Pregnant women experiencing adversity (≥2 of 10 risk factors) were recruited from 10 antenatal clinics across 2 states (Victoria, Tasmania) in Australia.

### Intervention

Mothers in the intervention arm were offered 25 nurse home visits (mean 23·2 home visits [SD 7·4, range 1–43] received) of 60–90 minutes, commencing antenatally and continuing until children's second birthdays.

### Primary and secondary outcomes measured

At 4 and 5 years, outcomes were assessed via parent interview and direct assessment of children's language and learning (receptive and expressive language, phonological awareness, attention, and executive function). Outcomes were compared between intervention

access/). Due to the type of ethical approval (for sensitive information), data sharing is subject to use for research processes and cannot be published on a public repository. This Data Availability statement has been accepted for a recently published PLOS ONE manuscript that drew on the same right@home dataset, written by some of the same authors as this submission (Bryson H, Mensah F, Price A...Goldfeld S. (2021) Clinical, financial and social impacts of COVID-19 and their associations with mental health for mothers and children experiencing adversity in Australia. PLOS ONE 16(9): e0257357; https://doi.org/10.1371/journal.pone.0257357).

**Funding:** This work is supported by the state governments of Victoria and Tasmania (n/a), the Ian Potter Foundation (n/a)(https://www.ianpotter.org.au/), Sabemo Trust (n/a) (https://sabemo.org.au/), Sidney Myer fund (n/a)(https://www.myerfoundation.org.au/), the Vincent Fairfax Family Foundation (n/a) (https://vfff.org.au/), and the National Health and Medical Research Council (NHMRC, 1079148) (https://www.nhmrc.gov.au/funding). The MCRI administered the research grant for the study and provided infrastructural support to its staff but played no role in the conduct or analysis of the trial. Research at the MCRI is supported by the Victorian Government's Operational Infrastructure Support Program. SG was supported by NHMRC Practitioner Fellowship (1155290). FM was supported by NHMRC Career Development Fellowship (1111160) (https://www.nhmrc.gov.au/funding) The funders had no role in study design, data collection and analysis, decision to publish, or preparation of the manuscript.

**Competing interests:** The "right@home" sustained nurse home visiting trial is a research collaboration between the Australian Research Alliance for Children and Youth (ARACY); the Translational Research and Social Innovation (TReSI) Group at Western Sydney University; and the Centre for Community Child Health (CCCH), which is a department of The Royal Children's Hospital and a research group of Murdoch Children's Research Institute. Ownership of the right@home implementation and support licence, which is purchased by Australian state governments for roll out, is shared between institutes.

**Abbreviations:** CFH, Child and Family Health; CI, confidence intervals; DASS, Depression Anxiety Stress Scales; ES, effect size; IPW, inverse probability weighting; MECSH, Maternal Early Childhood Sustained Home-visiting; MD, mean difference; MI, multiple imputation; NFP, Nurse Family Partnership; NHV, nurse home visiting; NIH, National Institutes of Health; OR, odds ratio;

and usual care arms (intention to treat) using adjusted regression with robust estimation to account for nurse/site. Missing data were addressed using multiple imputation and inverse probability weighting.

## Results

Of 722 women enrolled in the trial, 225 of 363 (62%) intervention and 201 of 359 (56%) usual care women provided data at 5 years. Estimated group differences showed an overall pattern favouring the intervention. Statistical evidence of benefits was found across child and maternal mental health and wellbeing, parenting and family relationships with effect sizes ranging 0·01–0·27.

## Conclusion

An Australian NHV program promoted longer-term family functioning and wellbeing for women experiencing adversity. NHV can offer an important component of a proportionate universal system that delivers support and intervention relative to need.

## Trial registration

2013–2016, registration ISRCTN89962120

## Introduction

Young children who experience sustained socioeconomic and psychosocial adversity (such as parent unemployment, poor parent mental health, family violence, and substance abuse) are at risk of wide-ranging health and developmental inequities [1, 2]. These persist throughout life, resulting in lower educational attainment, poorer health and lower income [3]. Inequities are costly for the individual and society, and considered a major public health issue [1]. Despite global policy interest [1], it is extremely challenging to implement programs that improve the health and developmental trajectories of children and families experiencing adversity. Indeed, in high-income countries with universal healthcare the 'inverse care law' persists, whereby families with the greatest need are least able to access sufficient services or those of highest quality [4]. Starfield's work into equitable primary care delivery [5], and Marmot's focus on proportionate universalism as a core aspect of system change [6], suggest that health systems for children should allocate resources equitably and according to risk, with appropriate intensity and flexibility to counter the social determinants of health.

Nurse home visiting (NHV) is one of the most widely implemented early interventions for addressing maternal and child health inequities. NHV provides targeted and intensive delivery of child and family health care, with the potential to overcome barriers to service access for families experiencing adversity [7]. Randomised controlled trials (RCTs) of NHV programs in high-income countries with universal healthcare include the Family Nurse Partnership (United Kingdom) [8], Pro Kind (Germany) [9] and VoorZoorg (Netherlands) [10]. These programs have predominately focussed on early outcomes (up to child age 3 years) and demonstrated wide-ranging benefits across parenting, the home environment, and child language and behaviour [7]. Notably, the diversity of program content and measurement, and relative scarcity of RCTs, have impeded meaningful meta-analysis [7]. Overall, the reported benefits of NHV tend to be variable and modest [7] which, at the population level, make sense when

PedsQL, Pediatric Quality of Life Inventory; RCT, randomised controlled trial; SDQ, Strengths and Difficulties Questionnaire; SEAPART, School Entry Alphabetic and Phonological Awareness Readiness Test; SEIFA, Socio-Economic Index for Areas; SD, standard deviation; SSIS, Social Skills Improvement System; US, United States.

considering the multi-dimensional nature of the programs, the variation in nurse practice, and the complex experience of adversity. In this context, the benefits of NHV have been noted as one of the few public health programs that can make a measurable difference for families and children in the earliest years of life [11].

Few NHV programs have been rigorously evaluated beyond children's third birthdays [7]. The most recent systematic review of NHV RCTs was conducted by Molloy et al. (2021) [7], and focused on outcomes measured from pregnancy to child age 5 years. Of 30 articles describing the results of seven NHV programs, only two reported evaluations to 4–5 years. Of these, the Nurse Family Partnership program improved children's language at 4 years [12], and the Minding the Baby program reduced children's externalising behaviours at 5 years [13]. In a separate RCT of the NFP program, Olds et al (2004) also reported improved child behaviour outcomes at 6 years [14]. However, these studies were conducted in the United States, where there is no universal health insurance and variability in the types of healthcare provision to serve as a comparator. As such, the findings are more limited in their application when considering the longer-term efficacy of NHV embedded in universal healthcare. In the context of high-quality child and family health (CFH) services, the comparative benefit of NHV may be different or less apparent, particularly over the longer-term. Only the German Pro Kind program has been evaluated for longer-term outcomes within a universal healthcare context [15]. At child age 7 years, Pro Kind demonstrated improvements to child behaviour and maternal mental health outcomes and reduced abusive parenting, with effect sizes (ES) ranging 0·19 to 0·25.

In Australia, a country with universal healthcare provision, "right@home" is the largest and only multi-site RCT of NHV [16]. Conducted in partnership with two Australian state governments, the right@home NHV program was offered to women from pregnancy (recruited in 2013–14) to their subsequent child's second birthday (2015–16). At the program's end, the trial demonstrated intervention benefits across the three primary outcome domains of parent care, responsivity and the home learning environment [17]. Evidence of benefits for maternal mental health and wellbeing also emerged at the 3-year-old follow-up, one year after the program ended [18]. Given the limited longer-term data on NHV effectiveness for countries with universal healthcare, we extended the right@home trial follow-up to child ages 4 and 5 years (2017–19) with the intention to directly assess children's language and learning outcomes up to the year before school-entry. Following the program logic underpinning the right@home program [19], we anticipated that the parenting-related benefits evident at 2 years, and the maternal mental health benefits evident at 3 years, would translate to emerging benefits in children's learning and development as well as sustained benefits for parenting and maternal mental health. We hypothesised that, at child ages 4 and 5 years, compared with families who received the universal CFH nursing service (usual care), families who received the NHV program (the intervention) would demonstrate improvements in children's language and learning, general and mental health, maternal mental health and wellbeing, parenting and family relationships.

## Methods

### Design and setting

RCT of NHV from pregnancy to child age 2 years, compared with the existing universal CFH service (usual care). As described in the published study Protocol (https://bmjopen.bmj.com/content/7/3/e013307) [16], right@home was conducted as a superiority trial with two parallel groups and a primary endpoint at child age 2 years. The current study focusses on the extended follow-up of child and maternal outcomes to child ages 4 and 5 years.

## Patient and public involvement

No patients or members of the public were involved in the design, conduct, reporting or dissemination of this project.

## Participants

Researchers recruited pregnant women attending antenatal clinics of 10 public maternity hospitals across the Australian states of Victoria and Tasmania from 30 April 2013 to 29 August 2014. Eligible women: (i) had due dates before 1 October 2014, (ii) were less than 37 weeks gestation at the time of recruitment, (iii) had sufficient English to complete face-to-face interviews, (iv) lived within travel boundaries specified by participating areas; and (v) had 2 or more of 10 risk factors identified at screening (young pregnancy; not living with another adult; no support in pregnancy; poor health; a long-term illness, health problem, or disability that limits daily activities; currently smokes; stress, anxiety or difficulty coping; low education; no person in the household currently earning an income; and never having had a job before) [16, 20]. Women were excluded if they: (i) were enrolled in an existing Tasmanian NHV program for 15-19-year-olds, (ii) did not comprehend the recruitment invitation (e.g. intellectual disability, or insufficient English), (iii) had no mechanism for contact, or (iv) experienced a critical event (e.g. termination of pregnancy, stillbirth or child death).

## Procedures, randomisation, and masking

Researchers identified eligible women in antenatal clinics via a screening survey and invited them into the RCT. Participants provided informed written consent for the RCT (initially to 2 years) and completed a home-based baseline interview with researchers. Participants were then randomised to control or intervention arms with a 1:1 allocation following a computer-generated schedule stratified by site (local government area) and parity (first-time parent versus those with children). An independent statistician generated the random allocation sequence. The research manager informed participants of their allocation. Research managerial staff, participants and intervention teams were aware of allocation. Researchers who conducted follow-up assessments were blinded to randomisation. Informed written consent to enrol in the extended follow-up to 5 years was invited by researchers conducting the 2-year home-based assessment. Researchers conducted follow-up assessments as home-based interviews with direct assessment when children were 4 and 5 years old.

## Program development

The right@home NHV program was custom-built for the Australian universal healthcare context, selecting features of existing programs, components and processes which had the greatest evidence for benefit and program effectiveness [19]. The resulting program was structured around the core Maternal (formerly Miller) Early Childhood Sustained Home-visiting (MECSH) framework and training [21], and bolstered by five evidence-based strategies for content (sleep, safety, nutrition, regulation, bonding/relationship) and two for the delivery process (video feedback and motivational interviewing strategies) [19, 22]. The program incorporated the 'curriculum' of usual care (the control), which offers broad-ranging support related to child health and development and parental wellbeing at child-age-based appointments. Implementation of the intervention was enabled using program logic that articulated improved longer-term (school entry) child and parent outcomes. Implementation also incorporated monitoring and feedback processes that ensured the program maintained fidelity while being adaptable to the real-world health system [19].

## Program delivery

Mothers in the intervention arm were offered 25 nurse home visits (mean 23·2 home visits [SD 7·4, range 1–43] received) of 60–90 minutes [22], commencing antenatally and delivered mostly by the same nurse trained in the right@home NHV model of care [14, 20]. Most intervention women (75·6%) also received one or more home visits by a social care practitioner (mean 1·7 visits [SD 2·0, range 0–15]) [22], who provided case management as needed. In contrast, the usual CFH service included a possible six (Tasmania) or nine (Victoria) free nurse consultations (of 20–40 minutes) up to 2 years (mean 7·6 consultations [SD 4·3] received) [17]. The usual CFH service comprised an initial home visit with subsequent appointments conducted in local clinics. The service in Victoria also included flexibility to offer additional home-based appointments in the first year postpartum depending on family need, as part of 'enhanced' (tiered) service provision.

## Outcome measures (Table 1)

Outcomes were identified *a priori*, across the domains of children's language and learning, general and mental health, parenting and family relationships, and maternal mental health and wellbeing. Given the complex nature of the right@home program, which was designed to improve multiple outcomes across child and parent domains, prioritising a single outcome (as is common in clinical RCTs) is likely to understate the effect of the intervention [23]. We therefore examined multiple outcomes, including direct assessment of child language and learning outcomes (receptive and expressive language, phonological awareness, attention and executive function). By examining the direction, magnitude, and confidence intervals of estimated effects for multiple outcomes, we aimed to comprehensively evaluate the evidence for treatment benefits [24].

## Statistical analyses

**Sample size.** The original RCT sample size was calculated to detect a minimum effect size (ES) of 0·3 SD in the continuous parent responsivity outcome at the 2-year primary endpoint, and was based on previous NHV trials [47, 48]. A target sample size of 714 participants was estimated to provide 80% power, with 5% significance level, accounting for clustering (non-independence of outcomes by the nurse providing NHV or site of usual care provision), and for 40% attrition. A final sample size of 722 participants was achieved. Retention over time was higher than anticipated and higher than comparable NHV trials [47]. At 2 years, 558/722 (77·3%) re-enrolled in extended follow-up. At 4 and 5 years, 64·4% and 59·0% participants provided data respectively (detailed in the Results). As the attrition at 5 years (41%) was that originally calculated for the 2-year primary endpoint, the available dataset retained the power of the original sample size calculation.

**Participant characteristics.** For women retained at 4 and 5 years, baseline characteristics of those in the intervention and usual care arm were compared using chi-square tests (categorical measures) and t-tests (continuous measures) to assess differences arising due to attrition. As the 5-year characteristics represent a longer follow-up period and a more comprehensive assessment, they are presented as primary results, and the 4-year characteristics are presented as supplementary results (S1 Table).

**Primary analyses.** The primary analysis was performed on an intention-to-treat basis including all the women who participated at 4 and 5 years, using a combination of multiple imputation (MI) and inverse probability weighting (IPW) to account for missing data amongst respondents and differential attrition, respectively.

**Table 1. Description of outcome measures assessed at 4 and 5 years.**

| Outcome measure | Description |
|---|---|
| *Child Language and Learning* | |
| Receptive and expressive language | Clinical Evaluation of Language Fundamentals Preschool Second Edition (CELF-P2) Australian Standardised Edition [25]. Direct assessment of child language skills across three subscales: Sentence Structure, Word Structure and Expressive Vocabulary, and a combined Core Language score. Subtest scores reported as age-specific normative scaled scores (m = 10, SD = 3) and Core Language score reported as standard score (m = 100, SD = 15). |
| Phonological awareness | School Entry Alphabetic and Phonological Awareness Readiness Test (SEAPART) [26]. Direct assessment of pre-literacy skills that are ideally mastered at the point of school entry. Subtest raw scores were assessed across 6 subscales: Syllable Clapping, Syllable Isolation, First Sound Identification, Letter Identification, Name Writing and Rhyme Detection. Total Score assessed as a combined score of all subtests, excluding Rhyme Detection. |
| Attention and executive function | Two subtests of the NIH Toolbox Early Childhood Cognition Battery (Ages 3–6 years) [27], assessing attention and executive function: Flanker Inhibitory Control & Attention Test and Dimensional Change Card Sort. Both administered to the child as direct assessments on the NIH Toolbox iPad application. Subtest scores examined as standardised age-corrected scores (mean = 100, SD = 15). |
| *Child General and Mental Health* | |
| Mental health and behaviour | 25-item Strengths and Difficulties Questionnaire (4– to 10-year-old version) [28, 29] assessing Total Difficulties and two domain scores: Internalising Difficulties (combined score of emotional and peer problems) and Externalising Difficulties (combined score of behaviour and attention/hyperactivity). Scores reversed so that higher mean scores indicate fewer problems. |
| Social Skills | 46-item Social Skills Improvement System [30] assessing social skills across 7 domains: Communication, Cooperation, Assertion, Responsibility, Empathy, Engagement and Self-Control. Total score assessed as a combined score of all 7 domains. Assessed at 5-year follow-up only. |
| Quality of life | 21-item Pediatric Quality of Life Inventory (PedsQL) [31] assessing general wellbeing using two subscales: Physical Functioning and Socioemotional Functioning. Higher scores indicate better wellbeing. |
| Stress (hair cortisol) | Hair cortisol as a measure of physiological stress response over the past 3 months. The hair sample is a minimum length of 3cm, with the total density of the sample equating to approximately half a pencil's width (30-50mg). Cortisol concentrations are log transformed and negativized so that higher scores indicate lower long-term stress [32]. |
| Global health | Single 5-point item ("poor" to "excellent") parent-report from the Short Form-6 (SF6) [33], dichotomised into poorer ("poor/fair/good") versus "very good/excellent" [34]. |
| No dental caries | Direct assessment of early signs of dental decay using 'lift the lip' screening to check the outer surface of the child's top front teeth [35]. Dichotomised into classifications of "healthy (no signs of dental caries)" versus "early/advanced signs of dental caries". |
| Not overweight or obese | Direct assessment of height (Invicta Stadiometer) and weight (Tanita HD-315 Digital Scales), used to calculate child age and gender specific BMI (weight/height2) z-scores, dichotomised into international classifications of weight status ("overweight/obese" versus "not overweight or obese (normal weight/underweight)"). |
| *Parenting and Family Relationships* | |
| Warm parenting | 6-item measure assessing parental warmth. Items rated on a 5-point scale ("never/ almost never" to "always/almost always"), drawn from LSAC [36]. |
| Hostile parenting | 5-item measure assessing parental hostility. Items rated on a 10-point scale ("not at all" to "all of the time"), drawn from LSAC [36]. |
| Parenting efficacy | 4-item Parenting Efficacy scale. Items rated on a 10-point scale ("Not at all how I feel" to "Exactly how I feel") drawn from LSAC, and a single 5-point Parenting Efficacy item assessing mother's feelings about herself as a parent ("Not very good" to "Very good") drawn from LSAC [36]. |

(*Continued*)

**Table 1.** (Continued)

| Outcome measure | Description |
|---|---|
| Parent-child closeness and conflict | 15-item short-form of the Child-Parent Relationship Scale (CPRS) [37, 38]. Parent-reported measure assessing views of their relationship with their child using two subscales: Conflict (scores reversed so that higher scores indicate less conflict) and Closeness (higher scores indicate greater closeness). |
| Regular meal times | Single 5-point item ("never" to "always"). Study-designed based on Sleep Well Be Well Regular Bedtime item [39]. |
| Regular bedtime | Single 5-point item ("never" to "always"), adapted from the "Sleep Well Be Well study [39]. |
| Regular bed routine | Single 5-point item ("never" to "always"), drawn from the "Sleep Well Be Well study [39]. |
| Child-parent relationship | Single 5-point item ("poor" to "excellent"), study designed based on the single Global Health item drawn from the self-reported Short Form-6 (SF6). |
| Intimate partner emotional abuse | 11-item Emotional Abuse subscale of the Composite Abuse Scales assessing emotionally abusive partner behaviour [40]. Dichotomised as a score of 3 or more indicating the presence of emotional abuse and a score less than 3 indicating no reported emotional abuse. |
| *Maternal Mental Health and Wellbeing* | |
| Mental health | Depression, Anxiety and Stress Scales [41]. 21-item measure, rated on a 4-point scale ("not at all" to "most of the time") assessing the negative emotional states of depression, anxiety and tension/stress. Three subscales (7 items each): Depression, Anxiety and Stress. DASS scores were reversed so that higher scores indicate better mental health, ranging from 0–21. Reversed DASS subscale scores were also dichotomised to reflect poorest mental health symptom severity (study-defined as lower 15% of scores according to population reference ranges [42]) versus better mental health (upper 85% of scores). |
| Wellbeing | Personal Wellbeing Index [43] 8 items assessing satisfaction with specific life domains, rated using a 10-point scale ("no satisfaction at all" to "completely satisfied") and summed for a total score ranging from 1–79 (total scores of 0 and 80 are excluded as response sets). |
| Quality of life | Assessment of Quality of Life– 8D (AQoL-8D) [44, 45]. 35-item measure assessing health related quality of life. Uses a weighting algorithm to provide a single overall utility-based quality of life measure on a scale of 0–1 (with 1 indicating a better quality of life). |
| Global health | Single 5-point item self-reported from the Short Form-6 (SF6) [33], dichotomised into poorer ("poor/fair/good") versus "very good/excellent" [34]. |
| Stress (hair cortisol) | See description for child stress (hair cortisol) above. |
| Currently employed | Maternal report of current employment (yes/no). |
| Currently studying | Maternal report of currently studying (yes/no). |
| Planning future study | Maternal report of planning future study (yes/no). |
| Self-efficacy | 3 items assessing mother's self-efficacy, drawn from the UK Millennium Cohort [46]. Each item reflected the presence versus absence of self-efficacy and were used to form a single dichotomous item reflecting 'any lack of self-efficacy' versus 'no lack of self-efficacy'. |
| Doesn't smoke | Maternal self-report of currently smoking (yes/no). |

Mean differences (MD) and odds ratios (OR) between intervention and control arms were estimated as measures of treatment effect for continuous and binary outcomes, respectively. Treatment effects were estimated using regression methods (linear or logistic models), adjusted for the stratification factors used during randomisation: parity and study site; and baseline covariates: family's Socio-Economic Index for Areas (SEIFA) score [49] indicating neighbourhood level disadvantage, maternal education, maternal age at child's birth, parity, antenatal risk, maternal self-efficacy, and maternal mental health, as well as child sex and age at the 4- and 5-year assessments. All regression analyses accounted for effects of nurse or site clustering using the robust cluster variance estimator [16].

It was expected that a high proportion of 4- and 5-year assessment data would be missing due to attrition. The primary analyses therefore used an approach that combines MI and IPW [50]. While MI is commonly implemented to deal with missing data, imputing large proportions of outcome data for those lost to follow-up can introduce bias. Using MI for participants with missed survey items, while using IPW to reweight the data to account for sample attrition, maintains the efficiency of MI while avoiding potential bias from imputing large proportions of data [50]. For both the 4- and 5-year analyses, the MI/IPW approach followed two steps. First, inverse probability weights were calculated for trial arms separately using the original samples of 363 program and 359 usual care mothers, adopting a logistic regression model predicting participation at follow-up (4 or 5 years), using baseline variables: marital status, employment status, nurse or site cluster, antenatal risk, maternal age at pregnancy, parity, site of recruitment, and mental health. Once inverse probability weights were obtained, only those who participated in the follow-up (4 or 5 years) were included in the next step with the weighting applied. In the second step, missing data of all participants at 4 or 5 years were imputed. MI models included all variables in the analysis models as well as all outcomes collected over previous years of the study. Multivariate normal regression by trial arm were used to impute 30 completed datasets, and each was then analysed applying the IPW derived in the first step.

Results in the Tables and Figures are reported as MD and standardised ES or with 95% confidence intervals (with all estimates integrating the MI and IPW missing data approaches). Complete case analyses, where all participant data for a given outcome are available, were undertaken to evaluate the sensitivity of the findings to sample attrition and missing data and are presented as supplementary results (S2 and S3 Tables). Data were analysed using Stata/IC v15 (Stata Corp, College Station, TX).

The right@home Expert Reference Group incorporated key stakeholders and experts to direct and advise on the technical design, implementation, and interpretation of the trial; no formal data monitoring committee was implemented. The trial is registered with an International Standard Randomised Controlled Trial Number (ISRCTN89962120; Trial status: complete). No other trials of the right@home intervention are ongoing.

### Ethical approval

right@home was approved by the Royal Children's Hospital Human Research Ethics Committee (HREC 32296), Australia.

### Role of the funding source

The funders played no role in the study design; in the collection, analysis, and interpretation of data; in the writing of the report; or the decision to submit the paper for publication.

## Results

Fig 1 shows that of the 722 women originally enrolled in the right@home trial, 465 (64·4%) provided data at 4 years (collected from 25 July 2017 to 3 January 2019) and 426 (59·0%) at 5 years (collected from 3 July 2018 to 14 January 2020). Table 2 shows that compared with women lost to follow-up at 5 years, those who participated in the intervention experienced less adversity at baseline (pregnancy) in terms of mental health, employment, ever having a drug problem or reporting family violence in the preceding year. Among those retained, baseline characteristics were similar between the intervention and usual care arms, although more children were female in the retained intervention families compared to usual care. S1 Table shows that the distribution of participant characteristics by follow-up status and trial arm was similar at 4 years.

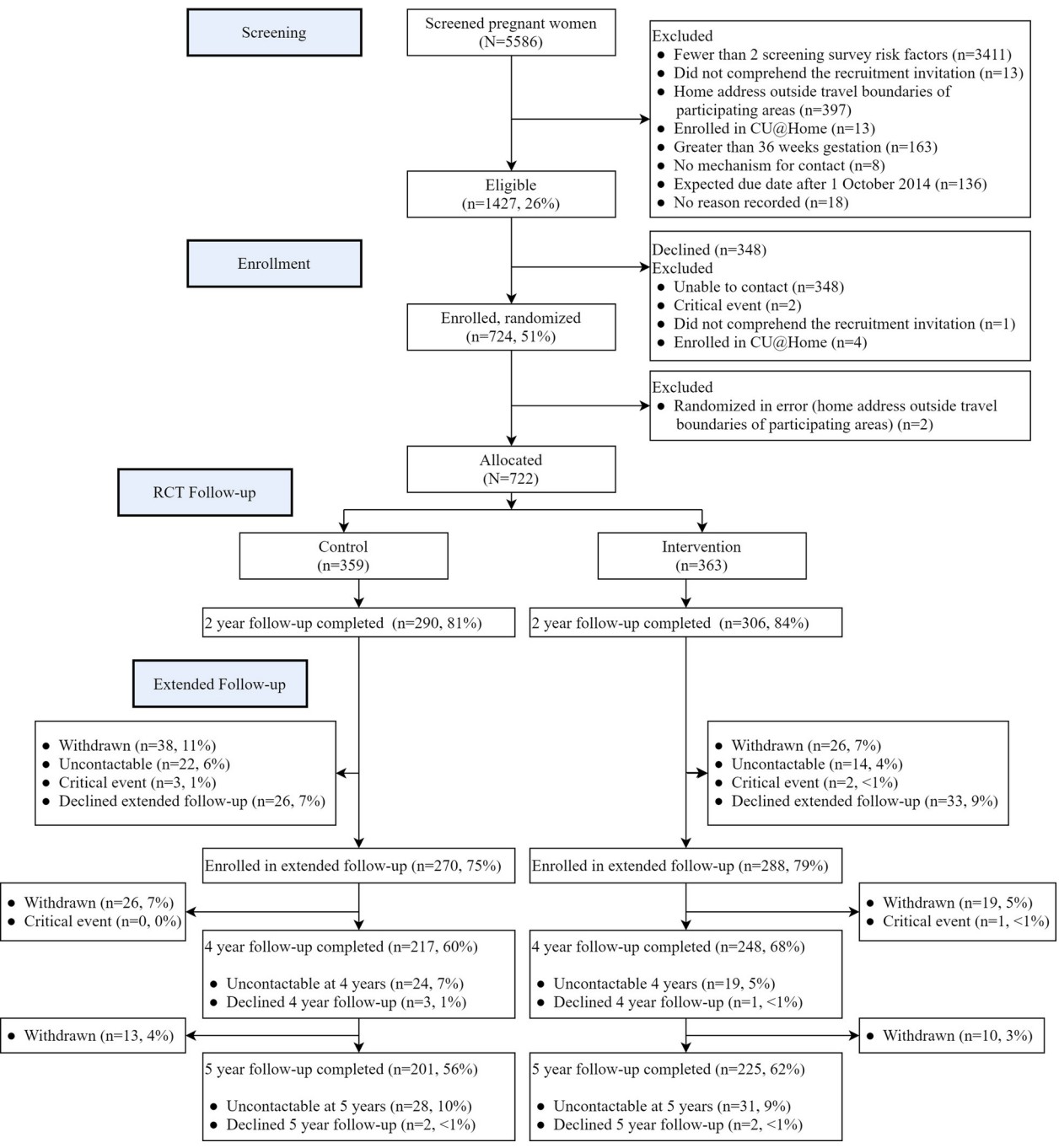

**Fig 1. Consort diagram.**

The adjusted regression analyses (with MI/IPW) show the direction of effects across child (Fig 2 and Table 3) and maternal (Fig 3 and Table 4) outcomes. The Figures present the direction, magnitude, and confidence of each estimate, allowing the overall evidence of treatment effect to be interpreted. Considered collectively, the estimated group differences for child and maternal outcomes favoured the intervention arm. The statistical evidence was stronger for parents than for children, and stronger at 5 years than 4 years. Effect sizes in favour of the intervention ranged from 0·01 to 0·27.

**Table 2. Baseline characteristics according to follow-up status (retained or lost to follow-up) at child age 5 years [a].**

| Baseline characteristics (pregnancy) | Intervention | | | Control | | |
|---|---|---|---|---|---|---|
| | Allocated (N = 363) | Retained (N = 225) | Lost (N = 138) | Allocated (N = 359) | Retained (N = 201) | Lost (N = 158) |
| *Mother* | | | | | | |
| Age (years), mean (SD) | 27·5 (6·1) | 27·9 (6·0) | 26·8 (6·1) | 27·8 (6·4) | 28·7 (6·4) | 26·7 (6·1) |
| DASS Depression, mean (SD) | 3·1 (3·6) | 3·1 (3·4) | 3·2 (4·0) | 2·9 (3·3) | 2·9 (3·2) | 2·8 (3·4) |
| DASS Anxiety, mean (SD) | 3·6 (3·5) | 3·6 (3·5) | 3·7 (3·4) | 3·4 (3·3) | 3·3 (3·1) | 3·6 (3·5) |
| DASS Stress, mean (SD) | 5·5 (4·3) | 5·4 (4·1) | 5·6 (4·5) | 5·4 (4·0) | 5·5 (4·0) | 5·2 (4·1) |
| DASS Depression, >85th percentile score | 64 (17·6) | 39 (17·3) | 25 (18·1) | 57 (15·9) | 27 (13·4) | 30 (19·0) |
| DASS Anxiety, >85th percentile score | 157 (43·3) | 94 (41·8) | 63 (45·7) | 149 (41·5) | 80 (39·8) | 69 (43·7) |
| DASS Stress, >85th percentile score | 73 (20·1) | 44 (19·6) | 29 (21·0) | 68 (18·9) | 39 (19·4) | 29 (18·4) |
| Education status | | | | | | |
| Did not complete high school | 80 (24·8) | 50 (24·5) | 30 (25·4) | 82 (25·3) | 46 (25·0) | 36 (25·7) |
| Completed high school / vocational training | 208 (64·6) | 130 (63·7) | 78 (66·1) | 207 (63·9) | 119 (64·7) | 88 (62·9) |
| Completed a university degree | 34 (10·6) | 24 (11·8) | 10 (8·5) | 35 (10·8) | 19 (10·3) | 16 (11·4) |
| Marital status | | | | | | |
| Single / not living with partner | 103 (28·4) | 65 (28·9) | 38 (27·5) | 92 (25·6) | 47 (23·4) | 45 (28·5) |
| Married / living with partner | 253 (69·7) | 157 (69·8) | 96 (69·6) | 260 (72·4) | 151 (75·1) | 109 (69·0) |
| Separated / divorced | 7 (1·9) | 3 (1·3) | 4 (2·9) | 7 (2·0) | 3 (1·5) | 4 (2·5) |
| Currently unemployed | 239 (65·8) | 138 (61·3) | 101 (73·2) | 239 (66·6) | 125 (62·2) | 114 (72·2) |
| Family income from benefit or pension | 159 (43·8) | 91 (40·4) | 68 (49·3) | 150 (41·8) | 81 (40·3) | 69 (43·7) |
| Ever had a drug problem | 51 (14·1) | 24 (10·7) | 27 (19·7) | 60 (16·9) | 23 (11·5) | 37 (23·7) |
| Experienced family violence in past year | 44 (12·2) | 22 (9·8) | 22 (16·2) | 41 (11·5) | 22 (11·0) | 19 (12·2) |
| Total adversity risk count (from screening), mean (SD) | 3·1 (1·3) | 3·0 (1·2) | 3·4 (1·4) | 3·2 (1·2) | 3·1 (1·2) | 3·3 (1·3) |
| *Child* | | | | | | |
| First child | 135 (37·2) | 88 (39·1) | 47 (35·1) | 131 (36·5) | 66 (32·8) | 65 (41·1) |
| Female | 191 (54·4) | 134 (59·6) | 57 (45·2) | 154 (44·6) | 97 (48·3) | 57 (39·6) |
| *Family* | | | | | | |
| SEIFA Index of Social Disadvantage Quintile | | | | | | |
| 1 (most disadvantaged) | 157 (44·5) | 97 (44·1) | 60 (45·1) | 139 (40·2) | 80 (41·5) | 59 (38·6) |
| 2 | 27 (7·7) | 15 (6·8) | 12 (9·0) | 30 (8·7) | 18 (9·3) | 12 (7·8) |
| 3 | 132 (37·4) | 91 (41·4) | 41 (30·8) | 132 (38·2) | 70 (36·3) | 62 (40·5) |
| 4 | 28 (7·9) | 11 (5·0) | 17 (12·8) | 32 (9·3) | 20 (10·4) | 12 (7·8) |
| 5 (least disadvantaged) | 9 (2·6) | 6 (2·7) | 3 (2·3) | 13 (3·8) | 5 (2·6) | 8 (5·2) |
| Language other than English | 29 (8·1) | 15 (6·8) | 14 (10·2) | 34 (9·7) | 16 (8·1) | 18 (11·8) |

[a] The 4-year characteristics are presented in S1 Table.

[b] p-value for chi-square tests (categorical measures) and t-tests (continuous measures) comparing those retained in the intervention and usual care arms.

All values are percentages, except where otherwise stated.

DASS = Depression, Anxiety, Stress Scale; SD = Standard Deviation; SEIFA = Socioeconomic Indexes for Areas Index of Relative Disadvantage

Range of Intervention N = 322–363, Control N = 324–359 due to missing data.

For children, there was statistical evidence of improved mental health outcomes at 5 years (reduced SDQ Externalising [ES: 0·14; 95% CI: -0·00 to 0·29] and reduced SDQ Total Score [ES: 0·20; 95% CI: 0·06 to 0·34]). Although the estimates favoured the intervention group, there was no statistical evidence of intervention effects for the directly assessed language and learning outcomes (receptive and expressive language, phonological awareness, attention, and executive function) at either time point.

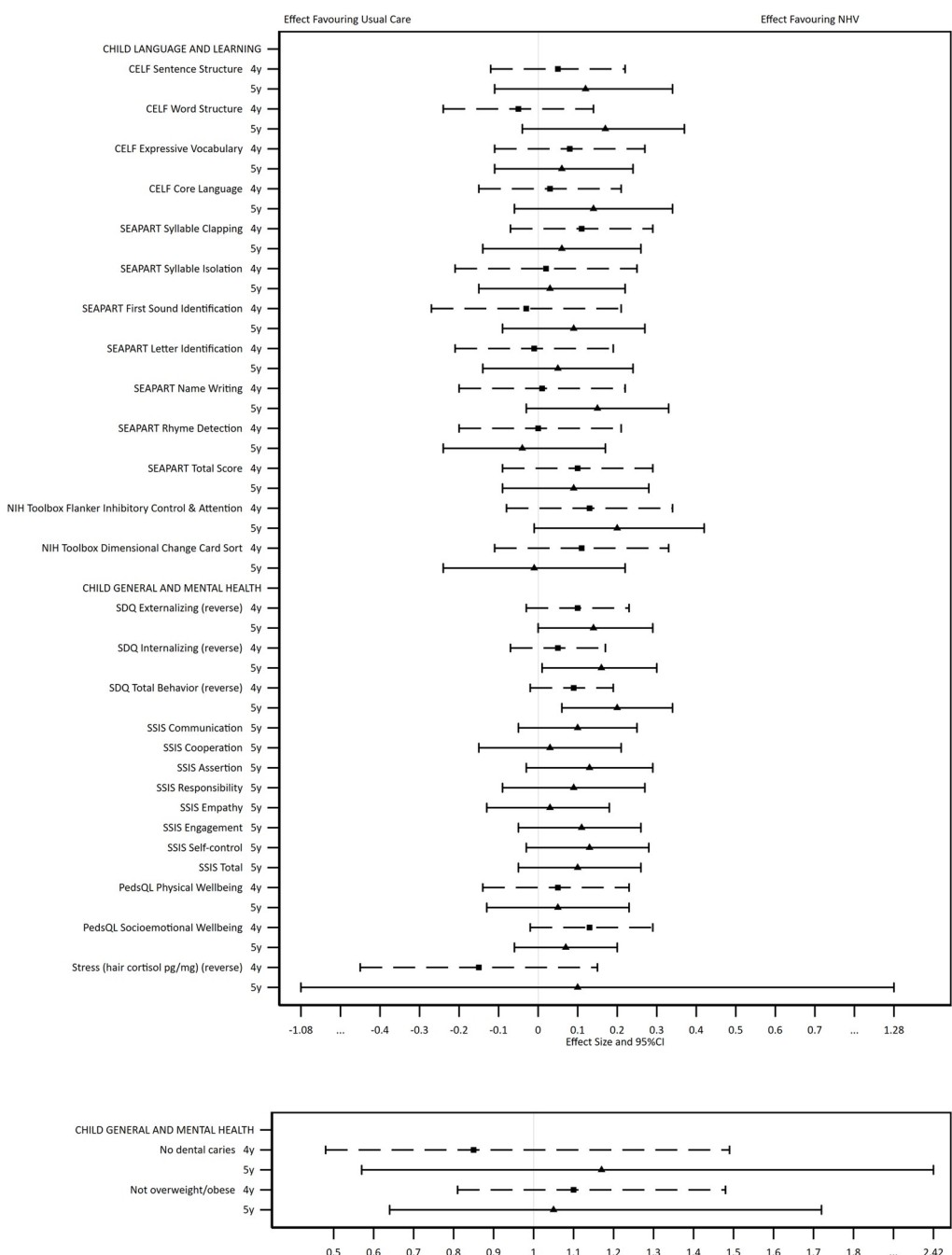

**Fig 2.** Results of adjusted regression analyses comparing the two trial arms on continuous (top box) and categorical (bottom box) child outcomes at 4 and 5 years, using multiply imputed and inverse probability weighted data.

There was evidence of intervention benefits across multiple outcomes of parenting at one or both time points (warmth at 5 years [ES: 0·18, 95% CI: 0·02 to 0·33], reduced hostility at 4 years [ES: 0·14, 95% CI: -0·01 to 0·28], efficacy at 5 years [ES: 0·21, 95% CI: 0·05 to 0·37], reduced parent-child conflict at 4 years [ES: 0·15, 95% CI: 0·01 to 0·29], regular mealtimes at 5 years [OR: 3·79; 95% CI: 1·81 to 7·92], regular bedtimes at 5 years [OR: 2·42; 95% CI: 1·43 to 4·10], and bedtime routine at 5 years [OR: 1·56; 95% CI: 1·08 to 2·26]). There was also evidence

**Table 3. Results of adjusted regression analyses comparing the two trial arms on child outcomes at age 4 and 5 years, using multiply imputed and inverse probability weighted data.**

| Outcome | Child age | Descriptive statistics | | Comparative statistic: I compared to C (95% CI) | | | | |
|---|---|---|---|---|---|---|---|---|
| | | Intervention (I) N = 248 (4y) N = 225 (5y) | Control (C) N = 217 (4y) N = 201 (5y) | Adjusted | | | Effect Size | 95% CI |
| | | | | Statistic [d] | 95% CI | p-value | | |
| *Child Language and Learning* | | Summary [a] | Summary [a] | | | | | |
| CELF Sentence Structure | 4y | 8·30 | 7·92 | 0·18 | -0·46; 0·83 | 0·56 | 0·05 | -0·12; 0·22 |
| | 5y | 9·19 | 8·65 | 0·43 | -0·39; 1·24 | 0·29 | 0·12 | -0·11; 0·34 |
| CELF Word Structure | 4y | 8·25 | 8·20 | -0·17 | -0·83; 0·49 | 0·59 | -0·05 | -0·24; 0·14 |
| | 5y | 8·88 | 8·16 | 0·64 | -0·14; 1·42 | 0·10 | 0·17 | -0·04; 0·37 |
| CELF Expressive Vocabulary | 4y | 8·75 | 8·37 | 0·26 | -0·38; 0·89 | 0·41 | 0·08 | -0·11; 0·27 |
| | 5y | 8·62 | 8·31 | 0·20 | -0·35; 0·76 | 0·44 | 0·06 | -0·11; 0·24 |
| CELF Core Language | 4y | 90·85 | 89·26 | 0·54 | -2·69; 3·76 | 0·73 | 0·03 | -0·15; 0·21 |
| | 5y | 93·58 | 90·49 | 2·51 | -1·05; 6·07 | 0·15 | 0·14 | -0·06; 0·34 |
| SEAPART Syllable Clapping | 4y | 5·24 | 4·82 | 0·40 | -0·26; 1·06 | 0·22 | 0·11 | -0·07; 0·29 |
| | 5y | 7·85 | 7·70 | 0·20 | -0·48; 0·88 | 0·54 | 0·06 | -0·14; 0·26 |
| SEAPART Syllable Isolation | 4y | 0·48 | 0·46 | 0·02 | -0·27; 0·32 | 0·89 | 0·02 | -0·21; 0·25 |
| | 5y | 2·61 | 2·43 | 0·11 | -0·50; 0·72 | 0·71 | 0·03 | -0·15; 0·22 |
| SEAPART First Sound Identification | 4y | 1·43 | 1·58 | -0·11 | -0·89; 0·68 | 0·78 | -0·03 | -0·27; 0·21 |
| | 5y | 5·28 | 4·86 | 0·46 | -0·45; 1·37 | 0·31 | 0·09 | -0·09; 0·27 |
| SEAPART Letter Identification | 4y | 1·25 | 1·28 | -0·02 | -0·56; 0·52 | 0·93 | -0·01 | -0·21; 0·19 |
| | 5y | 3·97 | 3·72 | 0·21 | -0·58; 1·00 | 0·57 | 0·05 | -0·14; 0·24 |
| SEAPART Name Writing | 4y | 0·76 | 0·71 | 0·02 | -0·35; 0·40 | 0·90 | 0·01 | -0·20; 0·22 |
| | 5y | 3·82 | 3·32 | 0·40 | -0·07; 0·87 | 0·09 | 0·15 | -0·03; 0·33 |
| SEAPART Rhyme Detection | 4y | 2·38 | 2·37 | 0·01 | -0·54; 0·56 | 0·97 | 0·00 | -0·20; 0·21 |
| | 5y | 2·53 | 2·57 | -0·06 | -0·37; 0·26 | 0·71 | -0·04 | -0·24; 0·17 |
| SEAPART Total Score | 4y | 9·90 | 9·01 | 0·91 | -0·83; 2·65 | 0·28 | 0·10 | -0·09; 0·29 |
| | 5y | 23·58 | 22·01 | 1·28 | -1·31; 3·86 | 0·30 | 0·09 | -0·09; 0·28 |
| NIH Toolbox Flanker Inhibitory Control & Attention | 4y | 95·61 | 92·99 | 2·19 | -1·41; 5·79 | 0·22 | 0·13 | -0·08; 0·34 |
| | 5y | 98·11 | 95·02 | 3·08 | -0·19; 6·35 | 0·06 | 0·20 | -0·01; 0·42 |
| NIH Toolbox Dimensional Change Card Sort | 4y | 97·02 | 94·90 | 1·63 | -1·62; 4·88 | 0·31 | 0·11 | -0·11; 0·33 |
| | 5y | 96·53 | 95·75 | -0·11 | -3·58; 3·36 | 0·95 | -0·01 | -0·24; 0·22 |
| *Child Health and Mental health* | | | | | | | | |
| SDQ Externalising (reverse) | 4y | 12·40 | 11·83 | 0·40 | -0·11; 0·92 | 0·12 | 0·10 | -0·03; 0·23 |
| | 5y | 13·63 | 12·80 | 0·57 | -0·01; 1·15 | 0·05 | 0·14 | -0·00; 0·29 |
| SDQ Internalising (reverse) | 4y | 16·19 | 16·01 | 0·14 | -0·21; 0·48 | 0·41 | 0·05 | -0·07; 0·17 |
| | 5y | 16·66 | 16·13 | 0·47 | 0·02; 0·92 | 0·04 | 0·16 | 0·01; 0·30 |
| SDQ Total Behaviour (reverse) | 4y | 28·58 | 27·89 | 0·49 | -0·13; 1·11 | 0·12 | 0·09 | -0·02; 0·19 |
| | 5y | 30·41 | 28·90 | 1·20 | 0·39; 2·01 | 0·01 | 0·20 | 0·06; 0·34 |
| SSIS Communication | 5y | 16·42 | 16·07 | 0·35 | -0·19; 0·90 | 0·19 | 0·10 | -0·05; 0·25 |
| SSIS Cooperation | 5y | 12·69 | 12·54 | 0·10 | -0·53; 0·73 | 0·74 | 0·03 | -0·15; 0·21 |
| SSIS Assertion | 5y | 15·03 | 14·71 | 0·44 | -0·12; 1·00 | 0·12 | 0·13 | -0·03; 0·29 |
| SSIS Responsibility | 5y | 12·32 | 11·92 | 0·32 | -0·33; 0·97 | 0·31 | 0·09 | -0·09; 0·27 |
| SSIS Empathy | 5y | 13·76 | 13·61 | 0·10 | -0·48; 0·67 | 0·73 | 0·03 | -0·13; 0·18 |
| SSIS Engagement | 5y | 15·97 | 15·58 | 0·43 | -0·18; 1·05 | 0·16 | 0·11 | -0·05; 0·26 |
| SSIS Self-control | 5y | 11·49 | 10·87 | 0·54 | -0·12; 1·21 | 0·10 | 0·13 | -0·03; 0·28 |

*(Continued)*

**Table 3.** (Continued)

| Outcome | Child age | Descriptive statistics | | Comparative statistic: I compared to C (95% CI) | | | | |
| | | Intervention (I) N = 248 (4y) N = 225 (5y) | Control (C) N = 217 (4y) N = 201 (5y) | Adjusted | | | Effect Size | 95% CI |
| | | | | Statistic [d] | 95% CI | p-value | | |
| SSIS Total | 5y | 97·70 | 95·31 | 2·21 | -1·11; 5·52 | 0·18 | 0·10 | -0·05; 0·26 |
| PedsQL Physical Wellbeing | 4y | 87·32 | 86·56 | 0·68 | -1·95; 3·31 | 0·60 | 0·05 | -0·14; 0·23 |
| | 5y | 88·00 | 87.16 | 0·66 | -1·65; 2·97 | 0·56 | 0·05 | -0·13; 0·23 |
| PedsQL Socioemotional Wellbeing | 4y | 82·19 | 79·97 | 1·87 | -0·29; 4·02 | 0·09 | 0·13 | -0·02; 0·29 |
| | 5y | 82·28 | 80·56 | 1·04 | -0·92; 2·99 | 0·28 | 0·07 | -0·06; 0·20 |
| Stress (hair cortisol, pg/mg [b]) | 4y | -1·81 | -1·62 | -0·21 | -0·64; 0·21 | 0·28 | -0·15 | -0·45; 0·15 |
| | 5y | -1·22 | -1·52 | 0·26 | -2·87; 3·39 | 0·68 | 0·10 | -1·08; 1·28 |
| No dental caries [c] | 4y | 78·16% | 79·37% | 0·85 | 0·48; 1·49 | 0·57 | N/A | N/A |
| | 5y | 83·93% | 82·55% | 1·17 | 0·57; 2·42 | 0·66 | N/A | N/A |
| Not overweight/obese [c] | 4y | 74·98% | 73·02% | 1·10 | 0·81; 1·48 | 0·55 | N/A | N/A |
| | 5y | 72·10% | 73·22% | 1·05 | 0·64; 1·72 | 0·84 | N/A | N/A |

I = Intervention; C = Control; CI = Confidence Interval; CELF = Clinical Evaluation of Language Fundamentals; NIH = National Institutes of Health;

PedsQL = Pediatric Quality of Life Inventory; SEAPART = School Entry Alphabetic and Phonological Awareness Readiness Test; SDQ = Strengths and Difficulties Questionnaire; SSIS = Social Skills Improvement System.

[a] Descriptive statistics are mean except where specified as dichotomous.

[b] Hair cortisol is log transformed and negativized, so that higher values indicate lower cortisol.

[c] Outcome is dichotomous (%), comparative statistics is odds ratio (OR).

[d] The comparative statistic is mean difference for continuous outcomes (intervention minus control) and odds ratio for dichotomous outcomes (the risk of outcome for those receiving the intervention compared with receiving usual care)

of intervention benefits to maternal mental health and wellbeing outcomes (reduced DASS stress symptoms at 5 years [ES: 0·12; 95% CI: -0·08 to 0·31, which was also reflected in the dichotomous stress variable], personal wellbeing at 4 and 5 years [ES: 0·17; 95% CI: 0·08 to 0·26, and ES: 0·27, 95% CI: 0·06 to 0·48], quality of life at 4 and 5 years [ES: 0·14, 95% CI: -0·02 to 0·30, and ES: 0·15, 95% CI: -0·01 to 0·32, which was also reflected in the dichotomous global health variable], and self-efficacy at 5 years [OR: 1·61, 95% CI: 1·13 to 2·31]). Intervention group mothers were more likely to report absence of emotional abuse from a partner at 5 years [OR: 1·72; 95% CI: 0·96 to 3·06]. The only outcome that showed evidence in favour of the usual care arm was that fewer intervention group mothers reported currently studying at 5 years [OR: 0·60; 95% CI: 0·40 to 0·92].

Similar associations were also identified in the complete cases analyses for child (S2 Table) and maternal (S3 Table) outcomes. In addition, there was evidence of benefit across the directly assessed child outcomes of receptive and expressive language (CELF Sentence Structure [ES: 0·21], Word Structure [ES: 0·15] and Core Language [ES: 0·17]), phonological awareness (SEAPART Total Score [ES: 0·21] and parent reported child mental health outcomes (reduced SDQ Internalising [ES: 0·20] and SSIS Self Control [ES: 0·20]) at 5 years in the complete cases analyses. Estimates were of similar magnitude to those in the MI/IPW analyses supporting a lack of bias introduced by attrition and missing data. Confidence intervals were generally narrower for complete cases reflecting the additional uncertainty introduced by using missing data techniques such as multiple imputation.

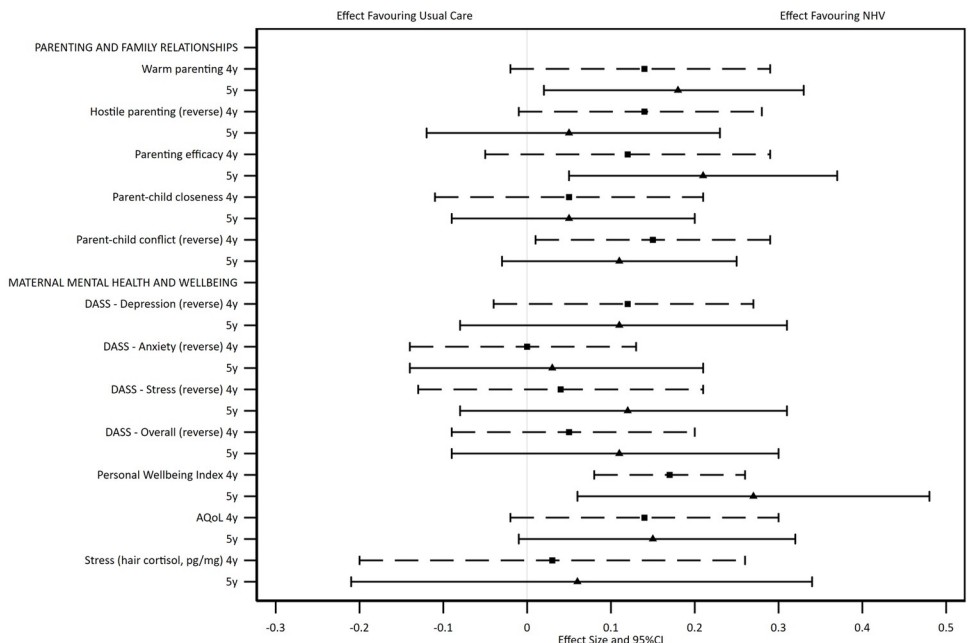

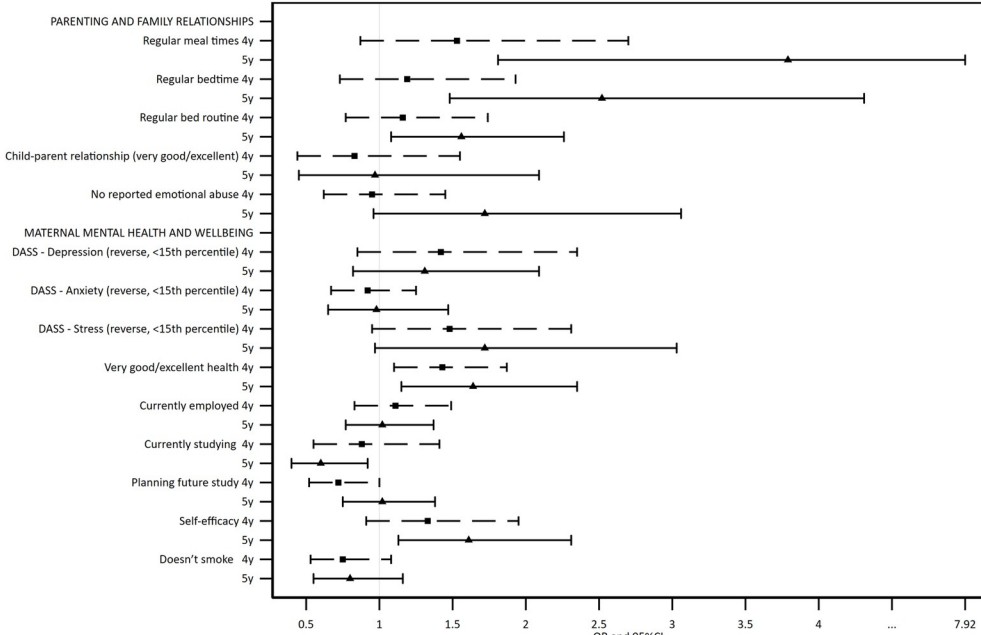

**Fig 3.** Results of adjusted regression analyses comparing the two trial arms on continuous (top box) and categorical (bottom box) maternal outcomes at 4 and 5 years, using multiply imputed and inverse probability weighted data.

## Discussion

This study reports the only longer-term evaluation of an Australian NHV trial designed to support women experiencing adversity from pregnancy to their children's second birthday and embedded in the universal CFH service. When children were 4 and 5 years old, which was two and three years after the intervention ended, there was evidence of modest benefits of the

**Table 4. Results of adjusted regression analyses comparing the two trial arms on maternal outcomes at child ages 4 and 5 years, using multiply imputed and inverse probability weighted data.**

| Outcome | Child age | Descriptive statistics | | Comparative statistic: I compared to C (95% CI) | | | | | |
|---|---|---|---|---|---|---|---|---|---|
| | | Intervention (I) N = 248 (4y) N = 225 (5y) | Control (C) N = 217 (4y) N = 201 (5y) | Adjusted | | | Effect Size | 95% CI | |
| | | | | Statistic [d] | 95% CI | p | | | |
| *Parenting and Family Relationships* | | Summary [a] | Summary [a] | | | | | | |
| Warm parenting | 4y | 4·66 | 4·58 | 0·07 | -0·01; 0·15 | 0·09 | 0·14 | -0·02; 0·29 | |
| | 5y | 4·73 | 4·64 | 0·08 | 0·01; 0·16 | 0·03 | 0·18 | 0.02; 0·33 | |
| Hostile parenting (reverse) | 4y | 7·94 | 7·73 | 0·23 | -0·01; 0·46 | 0·06 | 0·14 | -0·01; 0·28 | |
| | 5y | 7·73 | 7·65 | 0·09 | -0·21; 0·39 | 0·53 | 0·05 | -0·12; 0·23 | |
| Parenting efficacy | 4y | 8·10 | 7·91 | 0·19 | -0·07; 0·45 | 0·15 | 0·12 | -0·05; 0·29 | |
| | 5y | 8·33 | 7·96 | 0·34 | 0·08; 0·60 | 0·01 | 0·21 | 0·05; 0·37 | |
| Parent-child closeness | 4y | 33·29 | 33·11 | 0·17 | -0·34; 0·68 | 0·50 | 0·05 | -0·11; 0·21 | |
| | 5y | 33·21 | 33·09 | 0·16 | -0·27; 0·58 | 0·45 | 0·05 | -0·09; 0·20 | |
| Parent-child conflict (reverse) | 4y | 21·93 | 20·81 | 1·01 | 0·04; 1·98 | 0·04 | 0·15 | 0·01; 0·29 | |
| | 5y | 21·93 | 20·95 | 0·76 | -0·22; 1·75 | 0·12 | 0·11 | -0·03; 0·25 | |
| Regular meal times [b] | 4y | 91·83% | 88·15% | 1·53 | 0·87; 2·70 | 0·14 | N/A | N/A | |
| | 5y | 95·90% | 88·29% | 3·79 | 1·81; 7·92 | 0·00 | N/A | N/A | |
| Regular bedtime [b] | 4y | 82·92% | 79·47% | 1·19 | 0·73; 1·93 | 0·48 | N/A | N/A | |
| | 5y | 89·73% | 79·90% | 2·52 | 1·48; 4·31 | 0·001 | N/A | N/A | |
| Regular bed routine [b] | 4y | 79·91% | 76·38% | 1·16 | 0·77; 1·74 | 0·49 | N/A | N/A | |
| | 5y | 82·77% | 77·07% | 1·56 | 1·08; 2·26 | 0·02 | N/A | N/A | |
| Child-parent relationship (very good/excellent) [b] | 4y | 90·60% | 91·81% | 0·83 | 0·44; 1·55 | 0·55 | N/A | N/A | |
| | 5y | 92·34% | 91·76% | 0·97 | 0·45; 2·09 | 0·94 | N/A | N/A | |
| No reported intimate partner emotional abuse [b] | 4y | 66·98% | 67·89% | 0·95 | 0·62; 1·45 | 0·80 | N/A | N/A | |
| | 5y | 75·00% | 63·80% | 1·72 | 0·96; 3·06 | 0·07 | N/A | N/A | |
| *Maternal Mental Health and Wellbeing* | | | | | | | | | |
| DASS—Depression (reverse) | 4y | 18·15 | 17·62 | 0·48 | -0·16; 1·13 | 0·13 | 0·12 | -0·04; 0·27 | |
| | 5y | 18·18 | 17·67 | 0,41 | -0·31; 1·14 | 0·25 | 0·11 | -0·08; 0·31 | |
| DASS—Anxiety (reverse) | 4y | 18·33 | 18·30 | -0·01 | -0·49; 0·46 | 0·95 | -0·00 | -0·14; 0·13 | |
| | 5y | 18·23 | 17·96 | 0·12 | -0·51; 0·76 | 0·69 | 0·03 | -0·14; 0·21 | |
| DASS—Stress (reverse) | 4y | 15·76 | 15·56 | 0·18 | -0·55; 0·92 | 0·61 | 0·04 | -0·13; 0·21 | |
| | 5y | 15·74 | 15ᵃ13 | 0·50 | -0·35; 1·35 | 0·24 | 0·12 | -0·08; 0·31 | |
| DASS—Overall (reverse) | 4y | 52·23 | 51·50 | 0·61 | -1·12; 2·35 | 0·47 | 0·05 | -0·09; 0·20 | |
| | 5y | 52·21 | 50·70 | 1·15 | -0·98; 3·28 | 0·27 | 0·11 | -0·09; 0·30 | |
| Personal Wellbeing Index | 4y | 59·76 | 57·30 | 2·17 | 1·05; 3·30 | 0·001 | 0·17 | 0·08; 0·26 | |
| | 5y | 60·37 | 56·69 | 3·44 | 0·80; 6·08 | 0·01 | 0·27 | 0·06; 0·48 | |
| AQoL | 4y | 0·70 | 0·67 | 0·03 | -0·00; 0·06 | 0·09 | 0·14 | -0·02; 0·30 | |
| | 5y | 0·70 | 0·66 | 0·03 | -0·00; 0·06 | 0·07 | 0·15 | -0·01; 0·32 | |
| Stress (hair cortisol, pg/mg) [c] | 4y | -1·42 | -1·47 | 0·03 | -0·19; 0·24 | 0·79 | 0·03 | -0·20; 0·26 | |
| | 5y | -1·17 | -1·23 | 0·06 | -0·22; 0·34 | 0·63 | 0·06 | -0·21; 0·34 | |
| DASS—Depression (reverse, <15th percentile) [b] | 4y | 87·24% | 83·24% | 1·42 | 0·85; 2·35 | 0·18 | N/A | N/A | |
| | 5y | 83·54% | 79·88% | 1·31 | 0·82; 2·09 | 0·25 | N/A | N/A | |
| DASS—Anxiety (reverse, <15th percentile) [b] | 4y | 78·19% | 78·65% | 0·92 | 0·67; 1·25 | 0·58 | N/A | N/A | |
| | 5y | 75·76% | 75·85% | 0·98 | 0·65; 1·47 | 0·92 | N/A | N/A | |

*(Continued)*

**Table 4.** (Continued)

| Outcome | Child age | Descriptive statistics | | Comparative statistic: I compared to C (95% CI) | | | | |
| | | Intervention (I) N = 248 (4y) N = 225 (5y) | Control (C) N = 217 (4y) N = 201 (5y) | Adjusted | | | Effect Size | 95% CI |
| | | | | Statistic [d] | 95% CI | p | | |
| DASS—Stress (reverse, <15th percentile) [b] | 4y | 86·56% | 82·63% | 1·48 | 0·95; 2·31 | 0·09 | N/A | N/A |
| | 5y | 85·27% | 78·22% | 1·72 | 0·97; 3·03 | 0·06 | N/A | N/A |
| Very good/excellent health [b] | 4y | 30·85% | 23·86% | 1·43 | 1·10; 1·87 | 0·01 | N/A | N/A |
| | 5y | 31·01% | 21·65% | 1·64 | 1·15; 2·35 | 0·01 | N/A | N/A |
| Currently employed [b] | 4y | 40·46% | 36·81% | 1·11 | 0·83; 1·49 | 0·47 | N/A | N/A |
| | 5y | 43·54% | 41·86% | 1·02 | 0·77; 1·37 | 0·87 | N/A | N/A |
| Currently studying [b] | 4y | 16·06% | 17·73% | 0·88 | 0·55; 1·41 | 0·58 | N/A | N/A |
| | 5y | 15·16% | 21·26% | 0·60 | 0·40; 0·92 | 0·02 | N/A | N/A |
| Planning future study [b] | 4y | 50·10% | 58·66% | 0·72 | 0·52; 1·00 | 0·05 | N/A | N/A |
| | 5y | 56·84% | 58·80% | 1·02 | 0·75; 1·38 | 0·90 | N/A | N/A |
| Self-efficacy [b] | 4y | 81·54% | 76·35% | 1·33 | 0·91; 1·95 | 0·14 | N/A | N/A |
| | 5y | 78·43% | 69·84% | 1·61 | 1·13; 2·31 | 0·01 | N/A | N/A |
| Doesn't smoke [b] | 4y | 64·77% | 68·82% | 0·75 | 0·53; 1·08 | 0·12 | N/A | N/A |
| | 5y | 63·74% | 66·94% | 0·80 | 0·55; 1·16 | 0·24 | N/A | N/A |

I = Intervention; C = Control; CI = Confidence Interval; DASS = Depression Anxiety and Stress Scales; AQoL = Adult Quality of Life; NA = Not Applicable

[a] Descriptive statistics are mean except where specified as dichotomous.

[b] Outcome is dichotomous (%).

[c] Hair cortisol is log transformed and negativized, so that higher values indicate lower cortisol.

[d] The comparative statistic is mean difference for continuous outcomes (intervention minus control) and odds ratio for dichotomous outcomes (the risk of receiving the intervention compared with receiving usual care)

NHV intervention across multiple domains including children's mental health, parenting and family relationships, and maternal mental health and wellbeing. While there was no evidence of benefits in children's language or learning, there was a pattern across estimated group differences of improvements (albeit small) in favour of the intervention.

When considered with the earlier waves of follow-up, this study found emerging benefits for child behaviour (which were not evident when first assessed at 3 years) [51], and sustained benefits across parenting, family functioning and maternal mental health outcomes which increased at 5 years [18, 51]. At 5 years, there was also evidence of reduced intimate partner emotional abuse. This is notable since the intervention was a parenting intervention, rather than one that specifically targeted mental health or intimate partner abuse. Given the dearth of effective preventative interventions targeting inequitable outcomes, our research strengthens the evidence for NHV as an important part of a proportionate universal healthcare system.

The benefits identified at 4 and 5 years are consistent with the intervention focus and curriculum delivered by nurses [22]. As described previously [22], maternal mental and physical health and wellbeing support were the most frequently delivered program components, provided in over 85% of nurse visits. Discussions about relationships, including partnership issues and social support were also common, recorded for 50–70% of visits, with the topic of domestic violence recorded in a quarter of all visits [22]. Other areas of focus included care planning, caregiver aims and goals, infant feeding, growth and development, and housing and environment (discussed in at least half of the nurse home visits). The fidelity data align with the trial's program logic [19], which articulated that early parental benefits would generate child

developmental gains, which appear to be emerging at 5 years. These findings strengthen the international NHV evidence base by linking the direct inputs of the intervention with longer-term outcomes. For example, the emerging reduction in intimate partner emotional abuse (noting the substantial intervention input on relationships) is important given the lack of effective preventive interventions in this space [52].

Our findings also align with the small number of existing studies that have examined the longer-term effects of NHV programs. The recent 7-year-old follow-up of the Pro Kind program evaluated in Germany's universal healthcare setting also reported benefits for children's behaviour, reductions in adverse parenting and improved maternal mental health outcomes, but no evidence of benefits for children's school performance outcomes [15]. In our study, effect sizes for estimated group differences with statistical evidence ranged 0·12 to 0·27, and for Pro Kind ranged 0·19 to 0·25. While the US Nurse Family Partnership and Minding the Baby programs were evaluated in a different service context, they too reported benefits to child behaviour at 4 to 6 years with available effect sizes ranging from 0·06 to 0·17 [13, 14]. NFP additionally reported no impact on children's executive functioning at 4 years and mixed evidence for their academic outcomes (again with equivalent effect sizes) at 6 years [12, 14]. These similar outcomes were achieved with distinct programmatic inputs that may be important for policy scaling, such as differing foci in the curriculum and number of visits (e.g. right@home delivered an average of 23·2 home visits, compared with NFP which delivered an average of 7 antenatal and 26 postnatal visits, and Pro Kind which delivered an average of 32.7 home visits).

The strengths of this study include the rigorous, real-world, and prospective RCT design, and the comprehensive, longitudinal follow-up. Although loss-to-follow-up increased over time, we retained a relatively high proportion of participants (59% over a 6-year study duration) given the high levels of adversity and re-enrolment processes. In comparison, by their respective 2-year follow-up, the UK Building Blocks study retained 71% of their cohort for self-reported outcomes [8], and Pro Kind retained less than 50% [9]. Recognising that NHV trials are susceptible to participant attrition, which could introduce non-comparability between the trial arms, we applied the best practice methods of using a combined MI/IPW approach to address missing data [50]. Consistency between the complete case analysis and analysis with missing data accounted for, indicates that the intervention benefits evident in both sets of analyses are unlikely to be artefacts.

The balance of evidence across the outcomes (as illustrated in the forest plots) indicate beneficial outcomes overall. However, a limitation of the MI/IPW methods is that they introduce uncertainty into the estimates. As such, all estimates had wide confidence intervals and few outcomes reached statistical significance, meaning it is difficult to closely identify the specific effects on each given outcome. Missing data for all outcomes were also treated using the same approach, where some outcomes may have different reasons for missingness. For example, questions about whether mothers were experiencing intimate partner emotional abuse could only be administered when there were no other family members present, resulting in higher proportions of missing data than other measures. It is also important to consider why group differences were evident for outcomes reported by mothers but not for those that were directly assessed. Ongoing follow-up of the right@home cohort when children start school will enable comparison of teacher and parent reported data, and reveal whether developmental impacts emerge at a later age.

In thinking about the future of the NHV evidence base, we note that our limited sample size and power for detecting small group effects will continue to be an ongoing challenge, as it is for other studies. A substantially larger-scale evaluation would be needed to identify longer-term benefits more definitively. Lifetime cost-benefit modelling of NFP in the US shows that benefits accrue to participants and taxpayers over a child's lifetime, producing positive returns on

investment [51, 53], which is similarly beyond the scope of almost all research trials. While cost evaluations in universal healthcare settings are limited, our economic evaluation of right@home at child age 3 years demonstrated consistent findings; NHV involves high up-front costs that will be recouped if, and as, benefits emerge over time including those from early identification and intervention [51]. Given the demonstrated benefits of NHV, policy evaluations using designs such as stepped-wedge trials alongside program implementation should be prioritised into the future with a strong focus on implementation rather than smaller RCTs [54].

This study has demonstrated that it is possible to deliver a proportionate and targeted program within a universal CFH service that can provide important protective benefits to children and families experiencing adversity. Although modest effects delivered at scale can have a significant population impact, even the best outcomes of NHV highlight that a single intervention cannot fully redress inequities in health and development experienced by families. A sustainable and effective response is likely to require sustained intervention across the life course, identifying effective interventions and their ideal time point(s), duration, and intensity to maximise impact [55].

## Conclusion

Our findings suggest that the right@home NHV program offered sustained benefits to parenting, family functioning and maternal mental health and wellbeing for families experiencing adversity, when embedded into Australia's universal CFH service. Given the dearth of well tested preventive interventions, right@home offers a potentially effective option for a proportionate universal system that hopes to deliver support and intervention relative to need. Now, more than ever, as inequities likely increase in response to the COVID-19 pandemic, programs like right@home are needed to support families who could benefit most.

## Supporting information

**S1 Table. Baseline characteristics according to follow-up status (i.e., retained or lost to follow-up) at child age 4 years.**
(DOCX)

**S2 Table. Results of adjusted regression analyses comparing the two trial arms on child outcomes at age 4 and 5 years, using complete cases data.**
(DOCX)

**S3 Table. Results of adjusted regression analyses comparing the two trial arms on maternal outcomes at child ages 4 and 5 years, using complete cases data.**
(DOCX)

**S1 Checklist. CONSORT 2010 checklist of information to include when reporting a randomised trial\*.**
(PDF)

**S1 File.**
(PDF)

**S2 File.**
(PDF)

## Author Contributions

**Conceptualization:** Sharon Goldfeld, Fiona Mensah, Anna Price, Lisa Gold, Francesca Orsini, Penelope Dakin, Tracey Bruce.

**Data curation:** Francesca Orsini, Susan Perlen.

**Formal analysis:** Francesca Orsini.

**Funding acquisition:** Sharon Goldfeld, Hannah Bryson, Fiona Mensah, Anna Price, Lisa Gold, Penelope Dakin, Diana Harris, Lynn Kemp.

**Investigation:** Sharon Goldfeld, Hannah Bryson, Bridget Kenny, Susan Perlen, Tracey Bruce, Lynn Kemp.

**Methodology:** Sharon Goldfeld, Hannah Bryson, Fiona Mensah, Lisa Gold, Bridget Kenny, Susan Perlen, Shalika Bohingamu Mudiyanselage, Penelope Dakin, Tracey Bruce, Diana Harris, Lynn Kemp.

**Project administration:** Sharon Goldfeld, Hannah Bryson, Susan Perlen.

**Resources:** Sharon Goldfeld, Hannah Bryson, Penelope Dakin, Diana Harris, Lynn Kemp.

**Supervision:** Sharon Goldfeld, Hannah Bryson, Susan Perlen.

**Validation:** Francesca Orsini.

**Visualization:** Francesca Orsini.

**Writing – original draft:** Fiona Mensah, Anna Price.

**Writing – review & editing:** Sharon Goldfeld, Hannah Bryson, Fiona Mensah, Anna Price, Lisa Gold, Francesca Orsini, Bridget Kenny, Susan Perlen, Shalika Bohingamu Mudiyanselage, Penelope Dakin, Tracey Bruce, Diana Harris, Lynn Kemp.

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
