## [Decision Letter · Decision Letter 0]

9 Aug 2022

PONE-D-22-13241Nurse home visiting to improve child and maternal outcomes: 5-year follow-up of an Australian randomised controlled trial.PLOS ONE

Dear Dr. Goldfeld,

Thank you for submitting your manuscript to PLOS ONE. After careful consideration, we feel that it has merit but does not fully meet PLOS ONE’s publication criteria as it currently stands. Therefore, we invite you to submit a revised version of the manuscript that addresses the points raised during the review process.

There are several major problems spotted by the reviewers in the methodology ddescribed and in the way the results are interpreted. Please address all the issues.

We look forward to receiving your revised manuscript.

Kind regards,

Andrea Martinuzzi

Academic Editor

PLOS ONE

Journal Requirements:

2. Thank you for submitting your clinical trial to PLOS ONE and for providing the name of the registry and the registration number. The information in the registry entry suggests that your trial was registered after patient recruitment began. PLOS ONE strongly encourages authors to register all trials before recruiting the first participant in a study.

1) your reasons for your delay in registering this study (after enrolment of participants started);

2) confirmation that all related trials are registered by stating: “The authors confirm that all ongoing and related trials for this drug/intervention are registered”.

3. We note that you have stated that you will provide repository information for your data at acceptance. Should your manuscript be accepted for publication, we will hold it until you provide the relevant accession numbers or DOIs necessary to access your data. If you wish to make changes to your Data Availability statement, please describe these changes in your cover letter and we will update your Data Availability statement to reflect the information you provide

"This work is supported by the state governments of Victoria and Tasmania, the Ian Potter Foundation, Sabemo Trust, Sidney Myer fund, the Vincent Fairfax Family Foundation, and the National Health and Medical Research Council (NHMRC, 1079148). The MCRI administered the research grant for the study and provided infrastructural support to its staff but played no role in the conduct or analysis of the trial. Research at the MCRI is supported by the Victorian Government's Operational Infrastructure Support Program. SG was supported by NHMRC Practitioner Fellowship (1155290). FM was supported by NHMRC Career Development Fellowship (1111160)."

"This work is supported by the state governments of Victoria and Tasmania (n/a), the Ian Potter Foundation (n/a)(https://www.ianpotter.org.au/), Sabemo Trust (n/a) (https://sabemo.org.au/), Sidney Myer fund (n/a)(https://www.myerfoundation.org.au/), the Vincent Fairfax Family Foundation (n/a) (https://vfff.org.au/), and the National Health and Medical Research Council (NHMRC, 1079148) (https://www.nhmrc.gov.au/funding). The MCRI administered the research grant for the study and provided infrastructural support to its staff but played no role in the conduct or analysis of the trial. Research at the MCRI is supported by the Victorian Government's Operational Infrastructure Support Program. SG was supported by NHMRC Practitioner Fellowship (1155290). FM was supported by NHMRC Career Development Fellowship (1111160) (https://www.nhmrc.gov.au/funding) The funders had no role in study design, data collection and analysis, decision to publish, or preparation of the manuscript."

Reviewers' comments:

Reviewer's Responses to Questions

**Comments to the Author**

1. Is the manuscript technically sound, and do the data support the conclusions?

Reviewer #1: Yes

Reviewer #2: Partly

2. Has the statistical analysis been performed appropriately and rigorously? 

Reviewer #1: Yes

Reviewer #2: Yes

3. Have the authors made all data underlying the findings in their manuscript fully available?

Reviewer #1: Yes

Reviewer #2: No

4. Is the manuscript presented in an intelligible fashion and written in standard English?

Reviewer #1: Yes

Reviewer #2: Yes

5. Review Comments to the Author

Reviewer #1: Important note: This review pertains only to ‘statistical aspects’ of the study and so ‘clinical aspects’ [like medical importance, relevance of the study, ‘clinical significance and implication(s)’ of the whole study, etc.] are to be evaluated [should be assessed] separately/independently. Further please note that any ‘statistical review’ is generally done under the assumption that (such) study specific methodological [as well as execution] issues are perfectly taken care of by the investigator(s). This review is not an exception to that and so does not cover clinical aspects {however, seldom comments are made only if those issues are intimately / scientifically related & intermingle with ‘statistical aspects’ of the study}. Agreed that ‘statistical methods’ are used as just tools here, however, they are vital part of methodology [and so should be given due importance].

COMMENTS: It must be appreciated that the study is excellently planned and executed. There are hardly any issues, the only comment is: no reference is provided for [line 242] Socio-Economic Index for Areas (SEIFA).

However, I have two small suggestions (for kind consideration of authors) as follows:

• To provide a description of baseline characteristics is entirely reasonable (since it is clearly important in assessing to whom the results of the trial can be applied), however, statistical comparison of baseline characteristics [last ‘p-value’ column in Table 2] is not desirable at all [because even if P-value turns out to be significant (while comparing baseline characteristics despite random allocation), it is, by definition, a false positive] as you then are supposed to be testing ‘randomization’ then, which in any single trial may not balance all baseline characteristics [Participants were then randomized to control or intervention arms, lines 171-2] because ‘randomization’ is a sort of ‘insurance’ and not a guarantee scheme.

References:

1. Stuart J. Pocock, et al., ‘Subgroup analysis, covariate adjustment and baseline comparisons in clinical trial reporting: current practice and problems’, Statistics in medicine, 2002; 21:2917–2930 [Particularly page 2927]

2. Harrington D, et al., ‘New guidelines for statistical reporting in the journal’, N Engl J Med 2019;381:285-6

[Important message (indirectly/ultimately indicated) from these articles: Never do any comparison with respect to ‘baseline’ characteristics {by applying statistical significance test(s)}, when allocation is done randomly].

• Though the measures/tools used are appropriate {like Depression, Anxiety and Stress Scales, Personal Wellbeing Index, or Assessment of Quality of Life – 8D (AQoL-8D)} most of them yield data that are in [at the most] ‘ordinal’ level of measurement [and not in ratio level of measurement for sure {as the score two times higher does not indicate presence of that parameter/phenomenon as double (for example, a Visual Analogue Scales VAS score or say ‘depression’ score)}]. Then application of suitable non-parametric test(s) is/are indicated/advisable [even if distribution may be ‘Gaussian’ (i.e. normal)]. Agreed that there is/are no non-parametric test(s)/technique(s) available to be used as alternative in all situation(s) [suitable / most desired/applicable], but should be used whenever/wherever they are available. [Why not direct comparison and not comparison through regression which is indirect as regression techniques are not originally developed for comparison?]

Moreover, I would like to ask two questions :

• What is BMJ’s patient partnership strategy (line 153)?

• According to lines 226-7 [At 4 and 5 years, 64·4% and 59·0% participants provided data respectively] do not you think that drop-outs are too much {though its ‘retaining the power of the original sample size calculation & using a combination of multiple imputation (52) and inverse probability weighting (53) to account for missing data amongst respondents as per lines 235-237}?

This manuscript is alright otherwise. We can accept it after minor revision, in my opinion.

Reviewer #2: The authors present analysis of 4 and 5 year follow-up of a Nurse Home Visiting (NHV) RCT in Australia.

They present results on 29 child outcomes and 26 maternal outcomes. The abstract reports an "overall pattern favouring the intervention" with "statistical evidence of benefits" found across child and maternal mental health and well-being, parenting and family relationships with effect sizes from 0.01-0.27. The analytic strategy is sophisticated and in line with best practice.

1. Pg 12. describes original power calculations for a 2 year primary outcome at N= 714 with 722 randomized. Given attrition, it is not clear how power of the original sample size calculation was "retained" at 4-5 years given N=465 at 4 and 426 at 5?

2. The curious term "statistical evidence" means statistically significant I assume. There is a variety of statistical evidence presented across the 55 reported outcomes. Most confidence intervals are wide and compatible with negative, positive, and null effects. The authors focus on point estimates but there needs to be some recognition that most CIs are wide and that opens consideration of what we can learn from the study. The conclusion that this NHV promoted longer-term family functioning and wellbeing for women is not compatible with most of the statistical evidence. This should also include examination of differences between 4 and 5 year results - why are there effects at 4 and not 5 and vice versa? Otherwise it contributes to cherry picking those results that favour the story that the intervention 'works'. Most of the evidence presented including the Cis suggests it is very hard to know what the effects of this NHV are ages 4 and 5.

3. What is the relevance of stating that CI were 'generally narrower" for complete case? Of course they are, but what is this alluding to?

4. P 24 states that there was "emerging benefits' for reduced intimate partner violence. There was 1 percentage point difference at 4 and 11 points at 5 - does this mean it was "emerging"? Again this seems an overly optimistic interpretation. It would be worthwhile also stating what these same findings for intimate partner violence were at earlier ages in the previous publications.

5. P26 end 2 nd para - a statement is made about intervention results being evident over and above potential selection bias related to more women undertaking further study in the control group. I don't know how the authors can know this. If there was selection bias on this or other characteristics it's hard to know how that affects any particular outcome.

6. P24 Multiple imputation and IPW weighting for attrition is not a "limitation" because it introduces "uncertainty" into estimates. There are several sources of "uncertainty" in producing any point estimate. MI/IPW is the "best" approach we have given current knowledge so this extra "uncertainty" is real and should be reflected in consideration of what we can learn from these results, given attrition, which is a feature of all such RCTs in vulnerable populations like this. The limitation is potential bias introduced by attrition - MI and IPW are the best that can be done to try to account for that. If that is the case - MI/IPW is the best we have - then why show complete case analysis at all?

7. Some consideration of why self-report outcomes showed some differences but not for objective assessments needs to be included. That is not uncommon in the NHV literature.

6. PLOS authors have the option to publish the peer review history of their article (what does this mean?). If published, this will include your full peer review and any attached files.

Reviewer #1: No

Reviewer #2: No

---

## [Author Response · Author response to Decision Letter 0]

8 Sep 2022

Please see response to reviewers document attached.

---

## [Decision Letter · Decision Letter 1]

3 Nov 2022

Nurse home visiting to improve child and maternal outcomes: 5-year follow-up of an Australian randomised controlled trial.

PONE-D-22-13241R1

Dear Dr. Goldfeld,

We’re pleased to inform you that your manuscript has been judged scientifically suitable for publication and will be formally accepted for publication once it meets all outstanding technical requirements.

Kind regards,

Andrea Martinuzzi

Academic Editor

PLOS ONE

Additional Editor Comments (optional):

Reviewers' comments:

Reviewer's Responses to Questions

**Comments to the Author**

1. If the authors have adequately addressed your comments raised in a previous round of review and you feel that this manuscript is now acceptable for publication, you may indicate that here to bypass the “Comments to the Author” section, enter your conflict of interest statement in the “Confidential to Editor” section, and submit your "Accept" recommendation.

Reviewer #1: All comments have been addressed

Reviewer #2: (No Response)

2. Is the manuscript technically sound, and do the data support the conclusions?

Reviewer #1: (No Response)

Reviewer #2: (No Response)

3. Has the statistical analysis been performed appropriately and rigorously? 

Reviewer #1: (No Response)

Reviewer #2: (No Response)

4. Have the authors made all data underlying the findings in their manuscript fully available?

Reviewer #1: (No Response)

Reviewer #2: (No Response)

5. Is the manuscript presented in an intelligible fashion and written in standard English?

Reviewer #1: (No Response)

Reviewer #2: (No Response)

6. Review Comments to the Author

Reviewer #1: COMMENTS: Since all of the comments made on earlier draft are considered positively & attended, I recommend the acceptance. The manuscript now has achieved acceptable level, in my opinion.

However, I request authors to note that the article quoted [‘THE IMPORTANCE OF THE NORMALITY ASSUMPTION IN LARGE PUBLIC HEALTH DATA SETS’] is very nice, but I talked about ‘ordinal’ level of measurement and even said “[even if distribution may be ‘Gaussian’ (also called ‘normal’)”. Had not I?

Reviewer #2: (No Response)

7. PLOS authors have the option to publish the peer review history of their article (what does this mean?). If published, this will include your full peer review and any attached files.

Reviewer #1: No

Reviewer #2: No

---

## [Editor Report · Acceptance letter]

9 Nov 2022

PONE-D-22-13241R1 

Nurse home visiting to improve child and maternal outcomes: 5-year follow-up of an Australian randomised controlled trial. 

Dear Dr. Goldfeld:

I'm pleased to inform you that your manuscript has been deemed suitable for publication in PLOS ONE. Congratulations! Your manuscript is now with our production department. 

Kind regards, 

on behalf of

Dr. Andrea Martinuzzi 

Academic Editor

PLOS ONE